# Multimorbidity gender patterns in hospitalized elderly patients

**Pere Almagro**[1,2]*, **Ana Ponce**[1,2], **Shakeel Komal**[1,2], **Maria de la Asunción Villaverde**[1,2], **Cristina Castrillo**[1,2], **Gemma Grau**[1,2], **Lluis Simon**[1,2], **Alex de la Sierra**[1,2]

**1** Multimorbidity Unit, Internal Medicine Service, University Hospital Mutua de Terrassa, University of Barcelona, Terrassa, Spain, **2** Internal Medicine Department, University Hospital Mutua de Terrassa, University of Barcelona, Terrassa, Spain

* 19908pam@comb.cat

**Data Availability Statement:** All relevant data are deposited in: https://figshare.com/articles/Multimorbidity/11427000.

## Abstract

Patients with multimorbidity and complex health care needs are usually vulnerable elders with several concomitant advanced chronic diseases. Our research aim was to evaluate differences in patterns of multimorbidity by gender in this population and their possible prognostic implications, measured as in-hospital mortality, 1-month readmissions, and 1-year mortality. We focused on a cohort of elderly patients with well-established multimorbidity criteria admitted to a specific unit for chronic complex-care patients. Multimorbidity criteria, the Charlson, PROFUND and Barthel indexes, and the Pfeiffer test were collected prospectively during their stays. A total of 843 patients (49.2% men) were included, with a median age of 84 [interquartile range (IQR) 79–89] years. The women were older, with greater functional dependence [Barthel index: 40 (IQR:10–65) vs. 60 (IQR: 25–90)], showed more cognitive deterioration [Pfeiffer test: 5 (IQR:1–9) vs. 1 (0–6)], and had worse scores on the PROFUND index [15 (IQR:9–18) vs. 11.5 (IQR: 6–15)], all p <0.0001, while men had greater comorbidity measured with the Charlson index [5 (IQR: 3–7) vs. 4 (IQR: 3–6); p = 0.002]. In the multimorbidity criteria scale, heart failure, autoimmune diseases, dementia, and osteoarticular diseases were more frequent in women, while ischemic heart disease, chronic respiratory diseases, and neoplasms predominated in men. In the analysis of grouped patterns, neurological and osteoarticular diseases were more frequent in females, while respiratory and cancer predominated in males. We did not find gender differences for in-hospital mortality, 1-month readmissions, or 1-year mortality. In the multivariate analysis age, the Charlson, Barthel and PROFUND indexes, along with previous admissions, were independent predictors of 1-year mortality, while gender was non-significant. The Charlson and PROFUND indexes predicted mortality during follow-up more accurately in men than in women (AUC 0.70 vs. 0.57 and 0.74 vs. 0.62, respectively), with both p<0.001. In conclusion, our study shows differing patterns of multimorbidity by gender, with greater functional impairment in women and more comorbidity in men, although without differences in the prognosis. Moreover, some of these prognostic indicators had differing accuracy for the genders in predicting mortality.

**Funding:** The authors received no specific funding for this work.

**Competing interests:** The authors have declared that no competing interests exist.

## Introduction

Globally, from 1990 to 2017, life expectancy increased by 7.4 years, reaching 80 years in some developed countries [1]. The aging population is linked to a marked growth in the prevalence of elderly patients with multiple concomitant chronic diseases. Recent studies have shown that the mean of these chronic diseases is about five in the general population over 80 years of age, and increasing to eight in hospitalized patients [2,3].

This has led to the search for new concepts and terminologies to replace the classic definition of comorbidity–understood as a primary disease with other secondary associated conditions–given the difficulty in deciding which disease is the predominant one in an individual patient. Regarding these newly proposed concepts, several authors have suggested as more appropriate the term multimorbidity, defined as the presence of two or more concomitant long-term diseases in the same patient [4,5].

About 7% of the total European population suffers from multimorbidity, and this prevalence can reach 90% in people over 85 years of age [6,7]. Consistently across studies, elderly people and those with lower incomes are more likely to be affected by multimorbidity [7]. In some investigations, female gender is also related to greater multimorbidity but with a longer survival, in the so-called "male-female health-survival paradox" [8]. This paradox refers to the fact that the greater life expectancy in women is penalized by several chronic diseases that entail an increased burden of disabling physical impairment and functional dependence. Conversely, males are more susceptible to chronic disease conditions with a worse vital prognosis, such as cancer and ischemic heart disease [1].

Multimorbidity is associated with a lower quality of life, poorer prognosis, and an increase in health expenditure. In addition, a subgroup of those with multimorbidity is acknowledged to require more complex health care: frail elders with several concomitant chronic diseases, repeated hospitalizations, and frequent ambulatory health care requirements [5].

To date, multiple studies have investigated the impact, according to gender, of comorbidities on patients with a single chronic disease [9]. Similarly, several population-based reports have looked at gender differences in patients with multimorbidity [5,10]. However, the quality of evidence on gender differences in clinical practice with elderly patients hospitalized with multimorbidity is poor, and prospective studies evaluating the prognosis of these patients are lacking [7]. Recently, the guidelines for multimorbidity of the National Institute for Health and Care Excellence (NICE) highlighted the need for prospective studies in patients with well-defined multimorbidity criteria in order to evaluate their prognosis [11].

Our main objective was to prospectively analyze the characteristics and differences in gender patterns of multimorbidity in hospitalized elderly patients and the prognostic implications of these characteristics and differences in terms of readmissions and mortality.

## Methods

This was a prospective cohort study, evaluating all patients admitted to a hospital medical ward specialized in the care of multimorbidity patients in the University Hospital Mutua de Terrassa, from September 1, 2015, to December 31, 2016. The patients were included on the basis of the first admission in the unit, excluding subsequent readmissions. Only patients with two or more of the following criteria were included: age $\geq$ 75 years, b) $\geq$ 2 chronic diseases, defined by a multimorbidity scale or Charlson index, c) Barthel $\geq$ 75 points, and 4) Pfeiffer $\geq$ 3 points [12,13,14,15].

Briefly, the unit is dedicated to the prevention, detection, and treatment of complications of hospitalization, preserving physical capacity, individualizing the management plan, and coordinating the hospital discharge with outpatient care units [6]. Patient data are shared through

an electronic medical record with the primary care physician and nurses, and those specific outpatient units specialized in the care of chronic complex patients.

During the stay, medical and social variables were collected alongside those concerning domiciliary treatment. Multimorbidity was evaluated using the multimorbidity classification of the Spanish Department of Health that includes 15 chronic pathologies selected for relevant severity or impact on daily life activities [11]. "S1 Table". The number of chronic diseases was also collected with the Charlson index, the ESMI scale for other comorbid conditions not included in the Charlson index, and the PROFUND index [13,16,17]. "S2 and S3 Tables".

The PROFUND index is a multicomponent prognostic scale designed and validated for patients with multimorbidity, which includes variables such as age, presence of the main caregiver, dyspnea, delirium during admission, physical functional dependence, and the number of hospitalizations in the previous year. PROFUND scores range from 0 to 36 points [17]. Physical functional status was assessed with the Barthel index, while cognitive status was measured using the Pfeiffer test and, in some cases—for example, suspicion of dementia previously undiagnosed—complemented with the Mini-Mental Cognitive Examination in the Spanish version by Lobo et al. (MMCE) [18]. Delirium was diagnosed using the confusion assessment method, and dysphagia was evaluated by a speech therapist [19]. Additionally, patients were grouped into the most clinically relevant patterns of disease: a) metabolic diseases (chronic kidney disease, diabetes mellitus, hypertension, dyslipidemia, and obesity); b) cardiovascular diseases (ischemic heart disease, heart failure, arrhythmias, cerebrovascular disease, and peripheral vascular disease); c) respiratory diseases (chronic obstructive pulmonary disease, asthma, pulmonary fibrosis, and sleep apnea); d) neurological-psychiatric diseases (neurological motor diseases, dementia, depression, and anxiety); e) osteoarticular diseases (osteoarthritis, osteoporosis, and osteoporotic fractures); f) cancer (active neoplasm); and g) miscellanea (autoimmune disease, bowel disease, hepatopathy, anemia, and thromboembolic disease).

Mortality was evaluated in terms of its occurrence during index admission and at 1-year's follow-up, while hospital readmissions were evaluated at 30-days and 1 year after the hospital discharge. Variables associated with prognosis with respect to in-hospital mortality, readmissions, and mortality during follow-up were explored and their gender differences analyzed. The follow-up was done through telephone calls, contact with the primary care physician, and review of electronic medical records.

### Statistical analysis

Qualitative variables were expressed as absolute frequencies and percentages, while quantitative variables were summarized as mean and standard deviation (SD) in the case of normal distribution, or median and interquartile range. Comparison among means was made with the Student's t-test for independent samples and the non-parametric test (Mann-Whitney U) for variables not distributed normally. Either the x2 test or the Fisher exact test was used for the comparison of proportions. Hazard ratios and 95% confidence intervals for survival time were calculated with Cox regression models, and statistical significance was obtained with the long-rank test. Multivariate analysis for mortality was also calculated with Cox logistic regression analysis. Variables entered in the model were chosen based on univariant analysis results and clinical significance.

For validation purposes, we used the cumulative/dynamic area under the receiver operating characteristic curve (ROC) to express the ability of different variables to predict all-cause 1-year mortality, both globally and distributed by gender. Dynamic cumulative ROC curves were selected as they are considered the most appropriate method when the considered outcome (in our case mortality) is a time-dependent variable [20,21]. We used the nearest-neighbor estimator (NNE) proposed by Heagerty, Lumley, and Pepe to estimate the area under the

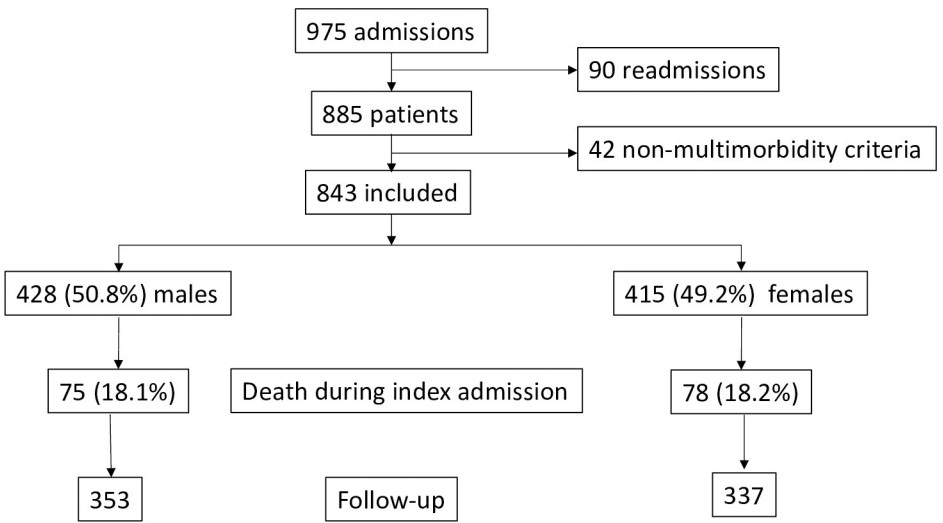

**Fig 1. Flowchart of participants.**

curve (AUC), and the naïve bootstrap procedure to estimate 95% confidence interval (95% CI) [20]. Detailed methodology is available elsewhere [21].

The STROBE check-list for cross-sectional studies is detailed in a supporting information file [22]. "S4 Table". The researchers read out explanations and provided them in written form, and informed consent was obtained from the patients, or, when this was impossible due to physical or cognitive impairment, from the caregivers. The signature was always made in the presence of the researcher and the patient. The Ethics and Clinical Trials Committee of the University Hospital Mutua de Terrassa approved the study.

## Results

Overall, 975 admissions of 885 patients were analyzed. Of these, 42 patients were excluded due to insufficient multimorbidity criteria. "Fig 1". The excluded patients were younger, with lower scores on the comorbidity scale and Pfeiffer and Charlson indexes, and with less functional dependence. No differences were found regarding sex and length of stay. "S5 Table".

### Differences by gender

The 843 patients included (49.2% men) had a median age of 84 (IQR: 79–89) years. As shown in Table 1, women were older, with greater functional dependence (Barthel index), more cognitive deterioration (Pfeiffer test), and worse scores on the PROFUND index (all p <0.0001). In contrast, men had greater comorbidity as measured by the Charlson index without age adjustment (p = 0.002). In the multimorbidity scale, heart failure, autoimmune diseases, dementia, and osteoarticular diseases were more frequent in women, while ischemic heart disease, chronic respiratory diseases, and neoplasms predominated in men. "Table 1".

In the analysis of grouped patterns, neurological and osteoarticular diseases were more frequent in females, while respiratory diseases and cancer predominated in males. "Table 2".

Similarly, the relation between the different grouped patterns of multimorbidity also differed by gender. The most frequent association in both genders was between cardiovascular and metabolic diseases. Cardioneurological pattern was the most frequent in women, while a cardiorespiratory pattern predominated in men. "Fig 2". Differences by gender in the diverse combinations of patterns are detailed in "S6 Table". "S1 Fig".

Table 1. Differences by gender in the studied population.

| Quantitative variables | Men (mean ± SD) | Women (mean ± SD) | Total (mean ± SD) | Men (median; IQR) | Women (median; IQR) | Total (median; IQR) | p |
|---|---|---|---|---|---|---|---|
| Age * | 80.5 (10.4) | 84.2 (8.8) | 82.4 (9.8) | 83 (75–87) | 85.5 (80–89) | 84 (79–89) | <0.0001 |
| Barthel* | 57.9 (34.5) | 41.1 (32.2) | 49.4 (34.4) | 60 (25–90) | 40 (10–65) | 50 (15–80) | <0.0001 |
| Pfeiffer* | 3.1 (3.7) | 5.3 (6.6) | 4.2 (5.5) | 1 (0–6) | 5.3 (6.6) | 2 (0–9) | <0.0001 |
| Mini Mental Cognitive Examination* | 25.5 (9.5) | 18.3 (11.6) | 21.7 (11.2) | 25 (20–32) | 18.3 (10–30) | 21.7 (14.8–32) | 0.001 |
| Charlson age-adjusted* | 8.7 (2.7) | 8.4 (2.3) | 8.6 (2.5) | 9 (7–10) | 8 (7–10) | 8 (7–10) | 0.1 |
| Charlson * | 5.4 (4.3) | 4.5 (2.1) | 4.9 (3.4) | 5 (3–7) | 4 (3–6) (2.1) | 4 (3–6) | 0.002 |
| PROFUND * | 11 (6.3) | 13.4 (5.9) | 12.2 (6.2) | 11.5 (6–15) | 15 (9–18) | 12 (8–18) | <0.0001 |
| Number domiciliary drugs * | 8.8 (4) | 9.2 (3.7) | 11 (6.3) | 8 (6–12) | 9 (7–12) | 9 (6–12) | 0.7 |
| Hospital stay (days) * | 10.7 (6.8) | 11.5 (10.4) | 11.1 (8.8) | 9 (6–14) | 8 (6–13) | 9 (6–13) | 0.7 |
| Number of hospitalizations *# | 2.6 (2.7) | 2.1 (2.4) | 2.3 (2.5) | 2 (0–4) | 1 (0–3) | 2 (0–4) | 0.02 |
| Qualitative variables | | | | number (%) | number (%) | number (%) | |
| Coexistence | | | | | | | |
| Alone | | | | 58 (14.1%) | 57 (13.3%) | 115 (13.7%) | 0.01 |
| Family | | | | 286 (69.4%) | 269 (62.9%) | 555 (66.1%) | |
| Profesioneal caregiver or nursing home | | | | 63 (15.3%) | 101 (23.6%) | 164 (19.5%) | |
| Others | | | | 5 (1.2%) | 1 (0.2%) | 6 (0.7%) | |
| Delirium | | | | 190 (46%) | 269 (63%) | 464 (55%) | <0.0001 |
| Dysphagia | | | | 190 (45%) | 241 (58%) | 430 (51%) | 0.002 |
| Comorbity_scale | | | | | | | |
| Hearth failure | | | | 224 (54%) | 266 (62.1%) | 490 (58.1%) | 0.01 |
| Ischemic heart disease | | | | 121 (29.2%) | 77 (18%) | 198 (23.5) | <0.0001 |
| Autoinmune | | | | 31 (7.5%) | 52 (12.1%) | 83 (9.8%) | 0.01 |
| Chronic Kidney Dis. | | | | 195 (47%) | 178 (41.6%) | 373 (44.2%) | 0.06 |
| Chronic Respiratory Dis. | | | | 227 (54.7%) | 181 (42.3%) | 408 (48.4%) | <0.0001 |
| Inflamatory bowel Dis. | | | | 9 (2.2%) | 14 (3.3%) | 23 (2.7%) | 0.2 |
| Chronic Liver Dis. | | | | 29 (7%) | 20 (4.7%) | 49 (5.8%) | 0.1 |
| Cerebrovascular | | | | 99 (23.9%) | 93 (21.7%) | 192 (22.8%) | 0.3 |
| Motor Neurological Dis. | | | | 49 (11.8%) | 51 (11.9%) | 100 (11.9%) | 0.5 |
| Dementia | | | | 141 (34%) | 198 (46.3%) | 339 (40.2%) | <0.0001 |
| Peripheral artery Dis. | | | | 78 (18.8%) | 30 (7%) | 108 (12.8%) | <0.0001 |
| Diabetes | | | | 114 (27.5%) | 114 (26.6%) | 228 (27%) | 0.04 |
| Chronic anemia | | | | 63 (15.2%) | 77 (18%) | 140 (16.6%) | 0.2 |
| Neoplasm | | | | 70 (16.9%) | 32 (7.5%) | 102 (12.1%) | 0.0001 |
| Chronic osteoarticular Dis. | | | | 41 (9.9%) | 128 (29.9%) | 169 (20%) | 0.0001 |
| Total comorbidity scale* | 3.6 (1.5) | 3.5 (1.4) | 3.5 (1.5) | 3.6 (2–5) | 3.5 (2–4) | 3.5 (2–5) | 0.7 |

*Non-normal distribution (Kolmogorov-Smirnoff. Analyses performed with Mann-Whitney U test.

# Number of hospitalizations in the previous year.

## In-hospital mortality

One hundred twenty-two patients (14.9%)—63 (15.2%) males and 59 (13.8%) females—died during their stay, without significant gender differences. Deceased patients were older [87 (IQR:82–91) vs. 84 (IQR:11.6); p <0.0001] years, with more comorbidity criteria measured with the Charlson index [5 (IQR:3–7) vs. 4 (IQR:3–6);p = 0.007], more functional dependence [Barthel 35 (IQR:10–69) vs. 55 (IQR:15–80); p = 0.0001], higher scores on PROFUND index

**Table 2. Gender patterns of multimorbidity.**

|  | men | women | total | p |
|---|---|---|---|---|
| Metabolic diseases | 363 (87.5%) | 379 (88.6%) | 742 (88%) | 0.3 |
| Cardiovascular diseases | 304 (73.3%) | 319 (74.5%) | 623 (73.9%) | 0.3 |
| Neurological diseases | 257 (61.9%) | 318 (74.3%) | 575 (68.2%) | <0.0001 |
| Respiratory diseases | 228 (54.9%) | 182 (42.5%) | 410 (48.6%) | <0.0001 |
| Osteoarticular diseases | 78 (18.8%) | 199 (46.5%) | 277 (32.9%) | <0.0001 |
| Neoplasm | 70 (16.9%) | 32 (7.5%) | 102 (12.1%) | <0.0001 |
| Miscellanea | 209 (50.4%) | 225 (52.6%) | 434 (51.5%) | 0.3 |

[15 (IQR:12–18) vs. 12 (IQR:7–16); p<0.0001], and greater cognitive impairment (Pfeiffer 6 (IQR: 2–9) vs. 2 (IQR:0–8); p<0.0001]. No differences were found with respect to the number of multimorbidity criteria, hospitalizations during the previous year, and number of chronic domiciliary drugs, between those deceased during admission and patients discharged alive. In the gender analysis, females who died during index hospitalization were older than males [87.3 (IQR:83–92) vs. 84.2 (IQR:80–89); p = 0.006], without differences in Charlson, Barthel, or PROFUND indexes, Pfeiffer test, or number of multimorbidity criteria. Only respiratory pattern in males was significantly associated with in-hospital mortality (p = 0.004). For the 122 patients deceased during the index stay, treatment ceiling was agreed upon with patients or caregivers in 110 (92.4%), and symptomatic treatment was initiated.

## Readmissions

Of the 721 patients discharged alive, eighty-six patients (11.9%)—45 (12.8%) males and 41 (11.1%) females—were readmitted within 1 month of discharge. To avoid immortal time bias we excluded patients who died after hospital discharge without readmission during the month after discharge. In this analysis the percentage of readmission was slightly higher (14.9% males and 12.9% females), without significant gender differences. Patients readmitted or dying during the month after discharge were older [85 (IQR: 79–89) vs. 83 (IQR: 78–88) years; p = 0.03] and had a higher number of hospital admissions in the previous year [2 (IQR:1–4) vs. 1 (IQR: 0–3); p = 0.02], as well as having more comorbidity [Charlson index: 5 (IQR:3–7) vs. 4 (IQR:3–6);p = 0.01], functional dependence [Barthel index: 40 (IQR: 10–75) vs. 55 (IQR: 25–85);p<0.0001], and cognitive impairment [Pfeiffer test: 4 (IQR:1–9) vs. 2 (IQR: 0–8); p = 0.007], as well as higher scores on the PROFUND index [14 (IQR:9–18) vs. 11 (IQR: 6–15); p<0.0001]. The leading causes of readmission were a new exacerbation of chronic disease in 14 cases (16.3%), infection related with the previous admission in 18 (20.9%), falls with traumatism in 7 (8.1%), dysphagia with aspiration in 28 (32.6%), and other causes in 19 (22.1%). Readmissions for a new exacerbation were more frequent in women, while dysphagia with aspiration was more frequent in men. "Fig 3"

During the year after discharge, 237 (67.3%) men and 248 (67.8%) women were readmitted (p = 0.4). The number of 1-year readmissions was similar for males and females: 1(IQR:0–3). "Table 3".

## 1-year mortality

One-year follow-up was available for 714 (99%) patients. Of these, 330 (46.2%) died; 152 (43.3%%) were men and 178 (49%) women (p = 0.1). "Fig 4". The median follow-up for deceased patients was 76 (IQR: 22–178) days. Predictors of 1-year mortality are detailed in Table 3. "Table 3". In the multivariate Cox regression analysis age, Charlson index, Barthel,

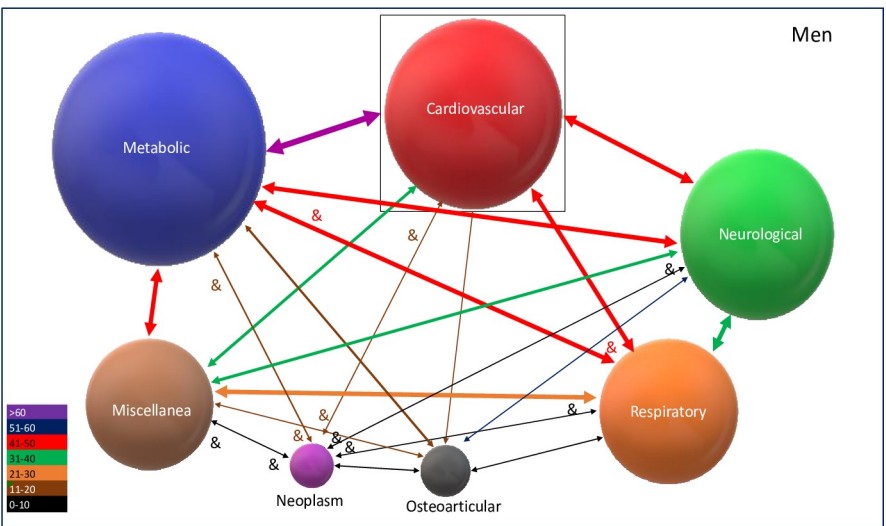

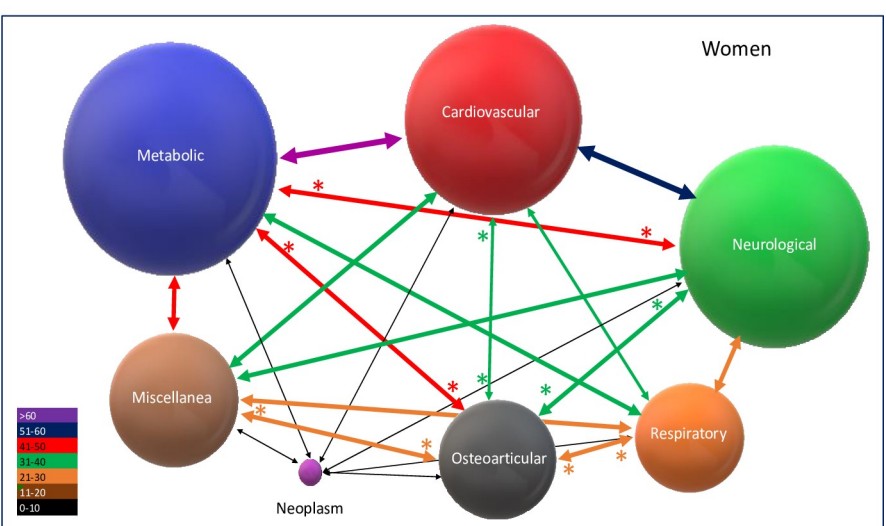

**Fig 2. The diameter of the spheres represents the prevalence in pattern percentage, while union lines express the frequency of the relationship.** Violet lines >60%, Blue lines 51–60%, Red lines 41–50%, Green lines 31–40%%, Orange lines 21–30, Brown lines 11–20%, Black lines <11%. * The association between patterns is statistically significantly higher in men than in women. & The association between patterns is statistically significantly higher in women than in men.

PROFUND, and previous admissions were predictors of 1-year mortality, while gender and the Pfeiffer test did not reach statistical significance in this model. "Table 4".

In the analysis of predictor models, performed with the cumulative/dynamic ROC curves, the AUC for 1-year mortality was 0.63, 0.67, and 0.68 for Charlson, Barthel, and PROFUND, respectively. "S2 Fig".

Regarding gender, PROFUND and Charlson index were better predictors of mortality in men than in women (AUC 0.74 for men vs. 0.62 for women in PROFUND index: p<0.001,

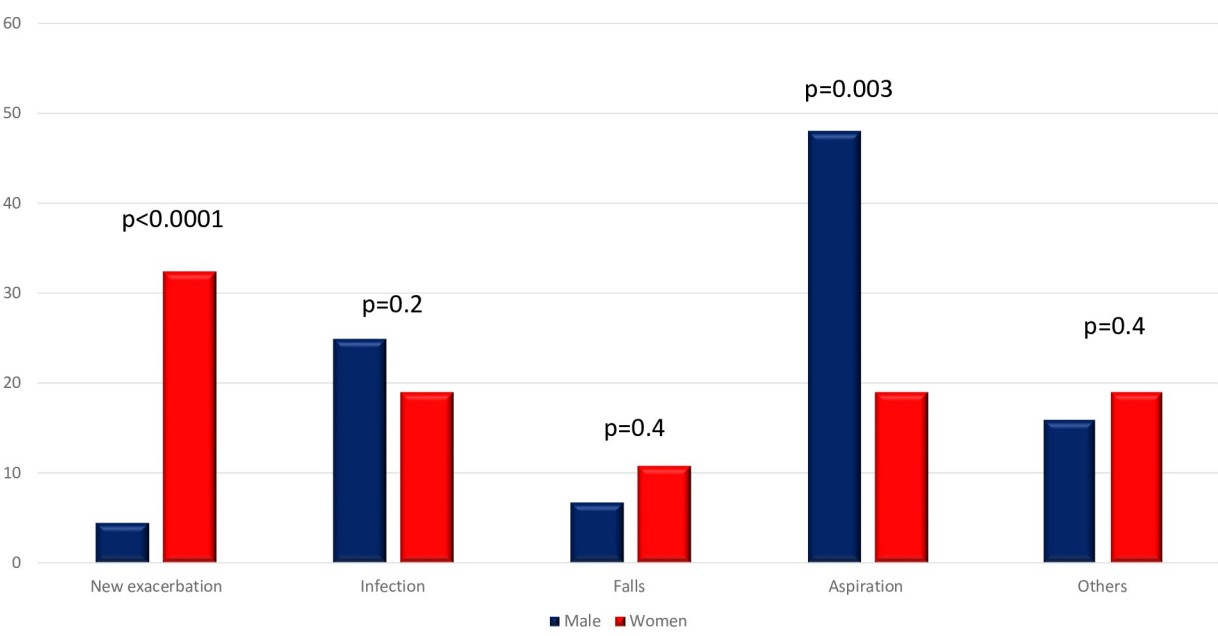

**Fig 3. Main cause and gender differences in 30-day readmissions.**

and 0.70 vs. 0.57 on the Charlson index for men and women: p<0.001). No differences were observed in the Barthel index according to gender. "Fig 5" Of the 167 patients with the highest quartiles in the PROFUND index (> 16 points), 63 (37.7%) were alive at the end of follow-up. Similarly, of the 148 patients with the worst scores on the Charlson index, 60 (40.5%) were alive at one year.

## Discussion

Our study demonstrates the presence of differences in gender patterns in multimorbidity patients hospitalized for decompensation of chronic diseases. Women were older, with greater functional dependence measured with the Barthel index and worse scores in cognitive evaluation. Conversely, males had a higher comorbidity load according to the Charlson index. In the multimorbidity scale, dementia, heart failure, and osteoarticular diseases predominated in

**Table 3. Univariate predictors of 1-year mortality.**

| Quantitative variables | Death (median;IQR) | Alive (median;IQR) | Hazard ratio | 95% C.I. | p |
|---|---|---|---|---|---|
| Age | 85 (80–89) | 82 (75–87) | 1.04 | 1.02–1.05 | <0.0001 |
| Barthel | 40 (10–65) | 65 (40–90) | 0.98 | 0.98–0.99 | <0.0001 |
| Pfeiffer | 5 (0–9) | 1 (0–6) | 1.1 | 1.07–1.14 | <0.0001 |
| Charlson age-adjusted | 9 (7–11) | 8 (6–10) | 1.15 | 1.11–1.2 | <0.0001 |
| Charlson | 5 (3–7) | 4 (3–6) | 1.12 | 1.07–1.17 | <0.0001 |
| Number of hospitalizations in the previous year | 2 (1–4) | 1 (0–3) | 1.06 | 1.02–1.09 | 0.003 |
| Number of multimorbidity criteria | 4 (3–5) | 3 (2–4) | 1.11 | 1.03–1.19 | 0.005 |
| Qualitative variables | | | | | |
| Gender (men) | 151 (46.6%) | 173 (53.4%) | 0.85 | 0.69–1.06 | 0.2 |
| Gender (women) | 198 (52.1%) | 182 (47.9%) | | | |
| Delirium | 150 (60.5%) | 115 (36.4%) | 2.25 | 1.73–2.87 | <0.0001 |
| Dysphagia | 109 (53.4%) | 70 (28%) | 2.34 | 1.78–3.08 | <0.0001 |

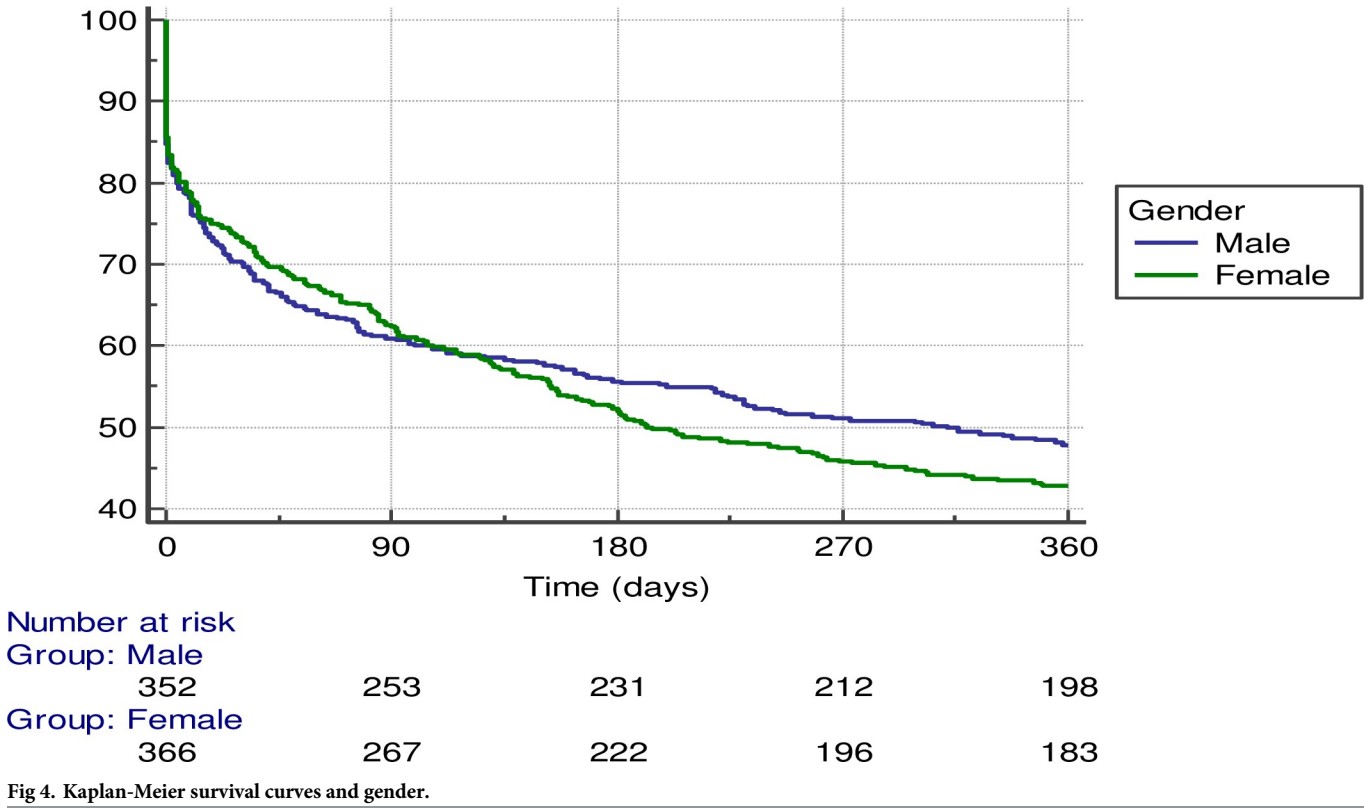

**Fig 4. Kaplan-Meier survival curves and gender.**

women, while in men ischemic heart disease, chronic respiratory diseases, and neoplasm were more prevalent. In the analysis of grouped patterns, neurological and osteoarticular diseases were more frequent in females, while respiratory diseases and cancer predominated in males.

To our knowledge, there are no previous studies that have explicitly explored gender patterns of multimorbidity in hospitalized older patients and their prognosis implications. We did not find differences by gender in terms of in-patient mortality or survival during the follow-up. This is in line with two Spanish studies performed in a population with similar characteristics to ours, although some previous reports have shown higher post-hospital mortality in males [10, 23,24,25].

In-patient mortality in our population (14.9%) was higher than that previously reported in other studies. According to the data from the Spanish National Statistics Institute in 2017,

**Table 4. Multivariate 1-year mortality analysis.**

|  | H.R. | 95% C.I. | p |
|---|---|---|---|
| Gender | 1.1 | 0.83–1.44 | 0.6 |
| Age | 1.02 | 1.01–1.04 | 0.03 |
| Charlson index* | 1.03 | 1.02–1.05 | <0.006 |
| Barthel index | 0.99 | 0.98–0.99 | 0.02 |
| PROFUND index | 1.06 | 1.03–1.1 | <0.0001 |
| Number hospitalizations in the previous year | 1.07 | 1.01–1.12 | 0.01 |
| Pfeiffer | 0.97 | 0.92–1.04 | 0.4 |

*Charlson index without age adjustment Please upload each supporting information element within separate files.

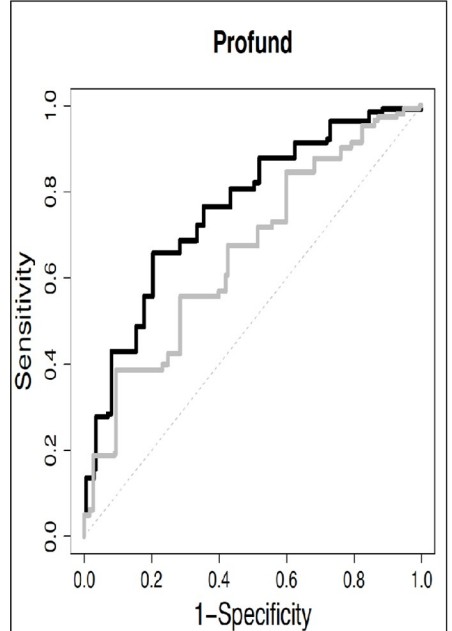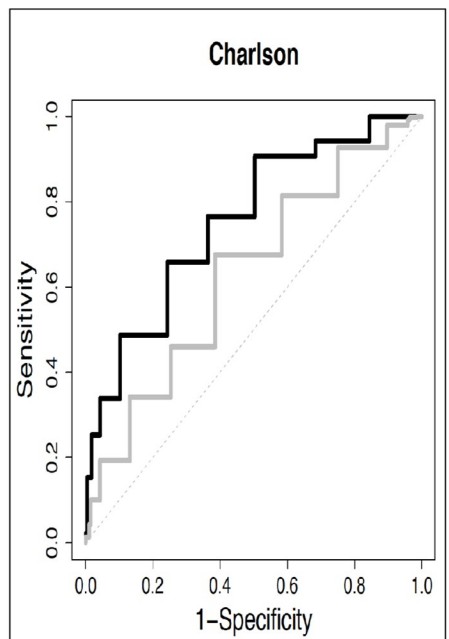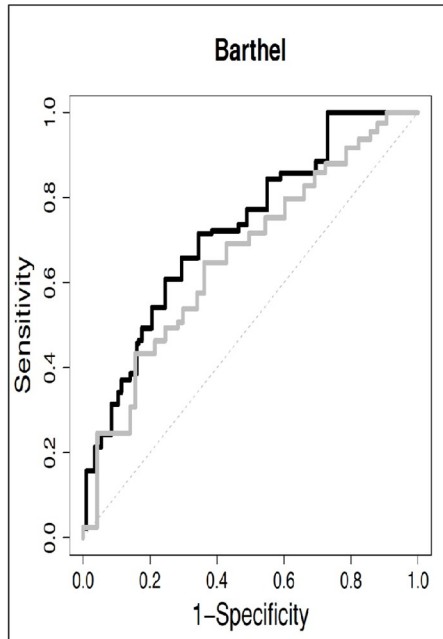

**Fig 5. Dynamic cumulative ROC curves and 1-year mortality for PROFUND, Charlson index, and Barthel scale.** Black = male. Gray = female.

internal medicine services in Spain issued a total of 905,659 hospital discharges, with a mortality rate of 9.24% [26]. However, the mean age of our population was nine years older, and the Charlson comorbidity index almost three times higher [27]. Our data on hospital mortality are similar to what was observed in studies performed with similar inclusion criteria, and we believe that the differences in mortality rates may simply reflect a sicker patient population [23,24,25]. Previous studies have shown that 70–80% of patients that die in hospital are frail, older patients with multiple comorbidities who were admitted through emergency rooms [28].

With many of these patients the main goal is not to prolong survival, but to reach agreement with the patient and caregivers on an individualized care plan and the careful treatment of distressful symptoms (dyspnea, confusion, pain, etc.) [29,30]. Furthermore, predicting the short-term prognosis of an individual patient is often difficult. A useful and well-valued approach for patients and physicians is to establish a therapeutic ceiling (for example, not to initiate cardiopulmonary resuscitation or to limit ventilator support to non-invasive ventilation) [31]. In more than ninety percent of patients dying during the index stay, the ceiling of treatment was previously agreed upon with the patients, or, if this was not possible due to the patient's condition, with their caregivers (usually the spouse or the children), and symptomatic or palliative treatment was started.

The observed percentage of readmissions in the month after discharge may be considered low. More importantly, the cause of readmission was in many cases related to secondary complications from the previous hospitalization. Although readmission is considered a criterion of quality of care, the Medicare data show that less than 40% of patients discharged with a diagnosis of heart failure, pneumonia, or chronic obstructive pulmonary disease are readmitted in the subsequent 30 days for the same cause, and fewer than one in four readmissions were deemed avoidable [32]. In the rest of the cases, readmissions were related to a vulnerable period after admission, called the post-hospital syndrome [33]. During this period, patients are more vulnerable to infections, falls, and reversible dysphagia [33,34]. Obviously, this vulnerable period is more relevant in previously frail elderly patients [25,35]. Classic studies have

shown that from 25% to 50% of elderly patients hospitalized for medical illness have a loss of independence in daily life activities, and that physical activity during hospital stays is severely reduced [36,37].

The figure for 1-year mortality observed in our study (46%) is in accordance with results published in other similar Spanish cohorts, where they ranged between 40% and 50%. In these studies as well, no differences were found in mortality by gender [23,24,38,39,40]. Additionally, our study confirms the reliability of the PROFUND and Charlson indexes in the prediction of 1-year mortality. The AUC observed for the PROFUND index (0.68) is nearly identical to the figures observed in these previous studies (0.7).

A relevant observation from our study, and one not previously explored, is that the predictive capacity of the Charlson and PROFUND indexes is clearly superior in males. That is, mortality is less predictable in women. In any case, although the performance of Charlson and PROFUND in our study may be considered good, we believe that no single predictive index should be used as an exclusive criterion to predict mortality in an individual multimorbid patient. For example, in the patients with worse scores on the PROFUND and Charlson index, 40% were alive at the end of follow-up; this percentage increased to 50% for the Charlson index in women. These data highlight the need for prior advanced decisions agreed upon with the patient and their caregivers. The unpredictability of the date of death should not involve subjecting the patient to unwanted aggressive care.

Our study has several limitations. First, it was conducted in a single center, with a specialized unit focused on the care of these patients, so perhaps the results cannot be extrapolated to other populations. However, the number of included patients was considerable, and their characteristics very similar to those observed in multicenter studies performed in the same geographical area. [17,23,24,38,39,40] Second, our study focuses on hospitalized elderly multimorbidity patients requiring complex health care, and their definition is not universally accepted. Currently, multimorbidity is defined as the presence of two or more chronic diseases in the same subject or the combination of one chronic illness with at least one other disease (acute or chronic) or bio-psychosocial factor [41]. This definition is too broad and fails to capture the complexity of some populations. For example, with this definition practically all community persons older than 85 years of age can be classified as multimorbid—since chronic diseases are closely linked to aging—and therefore the utility of this definition with this population is questionable [42]. More recently, some authors and health organizations have used different terms for patients such as complex chronic care or great health care needs. These patients are generally the frail elderly with multimorbidity, and we believe that this classification better describes our population [43]. Finally, other useful predictors of mortality in elderly multimorbid patients, such as the history of previous syncope, were not analyzed in our study [44].

In conclusion, our data confirm the differences in gender patterns of multimorbidity and the male-female health-survival paradox. Women were older, with greater functional dependence and greater prevalence of dementia and heart failure. Conversely, males had a greater load of global comorbidity measured with the Charlson index, and a greater predominance of ischemic heart disease, chronic respiratory diseases, and neoplasm.

## Supporting information

**S1 Table. Multimorbidity criteria Spanish Health Department**
(DOCX)

**S2 Table. Other comorbidities not included in the Charlson index (ESMI study).** History or treatment
(DOCX)

**S3 Table. PROFUND index**
(DOCX)

**S4 Table. STROBE Statement—Checklist of items that should be included in reports of cohort studies.**
(DOCX)

**S5 Table. Differences between included and excluded patients.**
(DOCX)

**S6 Table. Gender differences in combination patterns of multimorbidity.**
(DOCX)

**S1 Fig. Correlogram of different combinations of multimorbidity, based on Spearman correlation test.** The colors of cells range from red (highest correlation coefficient) to green (lowest correlation coefficient).
(DOCX)

**S2 Fig. Cumulative-dynamic ROC curves for 1-year mortality and PROFUND, Charlson, and Barthel indexes.**
(DOCX)

## Acknowledgments

The authors thank Tom Yohannan for medical writing services.

## Author Contributions

**Conceptualization:** Pere Almagro.

**Data curation:** Pere Almagro.

**Formal analysis:** Pere Almagro.

**Investigation:** Ana Ponce, Shakeel Komal, Maria de la Asunción Villaverde, Cristina Castrillo, Gemma Grau, Lluis Simon.

**Methodology:** Pere Almagro.

**Supervision:** Alex de la Sierra.

**Writing – original draft:** Pere Almagro.

**Writing – review & editing:** Alex de la Sierra.

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
