## [Decision Letter · Decision Letter 0]

21 Oct 2019

PONE-D-19-26678

MULTIMORBIDITY GENDER PATTERNS IN HOSPITALIZED ELDERLY PATIENTS.

PLOS ONE

Dear DR ALMAGRO,

Thank you for submitting your manuscript to PLOS ONE. After careful consideration, we feel that it has merit but does not fully meet PLOS ONE’s publication criteria as it currently stands. Therefore, we invite you to submit a revised version of the manuscript that addresses the points raised during the review process.

We would appreciate receiving your revised manuscript by Dec 05 2019 11:59PM. To enhance the reproducibility of your results, we recommend that if applicable you deposit your laboratory protocols in protocols.io, where a protocol can be assigned its own identifier (DOI) such that it can be cited independently in the future. For instructions see: http://journals.plos.org/plosone/s/submission-guidelines#loc-laboratory-protocols

We look forward to receiving your revised manuscript.

Kind regards,

Pasquale Abete

Academic Editor

PLOS ONE

Journal Requirements:

2. Please amend the manuscript submission data (via Edit Submission) to include author de la Asunción Villaverde.

Additional Editor Comments (if provided):

The authors evaluate differences in patterns of multimorbidity by gender in this population and their possible prognostic implications defined as in-hospital mortality, 1-month readmissions, and 1- year mortality in 843 patients with well-established multimorbidity criteria admitted to a specific unit for chronic complex care patients. Multimorbidity criteria, Charlson, PROFUND and Barthel indexes, and Pfeiffer test were collected prospectively during admission. Women were older, with greater functional dependence , more cognitive deterioration ], and worse scores on the PROFUND In the multimorbidity criteria scale, heart failure, autoimmune diseases, dementia, and osteoarticular diseases were more frequent in women, while ischemic heart disease, chronic respiratory diseases, and neoplasms predominated in men. In the analysis of grouped patterns, neurological and osteoarticular diseases were more frequent in females, while respiratory and cancer predominated in males. We did not find gender differences for in-hospital mortality, 1- month readmissions, or 1-year mortality. In the multivariate analysis age, Charlson, Barthel and PROFUND indexes, alongside previous admissions, were independent predictors of 1-year mortality, while gender was non-significant.

The manuscript is very interesting. I have only a minor concern. No mention about the prevalence and incidence of syncope that is well known to influence mortality and disability in older adults. Do you have any data about this matter. Please see and discuss Ungar A et al. Two-year morbidity and mortality in elderly patients with syncope. Age Ageing. 2011 Nov;40(6):696-702.

Reviewers' comments:

Reviewer's Responses to Questions

**Comments to the Author**

1. Is the manuscript technically sound, and do the data support the conclusions?

Reviewer #1: Yes

Reviewer #2: Yes

2. Has the statistical analysis been performed appropriately and rigorously? 

Reviewer #1: Yes

Reviewer #2: Yes

3. Have the authors made all data underlying the findings in their manuscript fully available?

Reviewer #1: Yes

Reviewer #2: Yes

4. Is the manuscript presented in an intelligible fashion and written in standard English?

Reviewer #1: Yes

Reviewer #2: Yes

5. Review Comments to the Author

Reviewer #1: In the manuscript entitled Multimorbidity gender patterns in hospitalized elderly patients, Almagro P. and co-authors report on the assessment of differences in patterns of multimorbidity by gender in a cohort of elderly patients and their possible prognostic implications defined as in-hospital mortality, 1-month readmissions, and 1-year mortality. To this aim Authors focused on a cohort of 843 elderly patients in which, based on the multimorbidity criteria scale, heart failure, autoimmune diseases, dementia, and osteoarticular diseases were more frequent in women, and ischemic heart disease, chronic respiratory diseases, and neoplasms predominated in men. Then the Authors grouped multimorbidity patterns and found that neurological and osteoarticular diseases were more frequent in females, while respiratory and cancer predominated in males but didn’t find gender-related differences for in-hospital mortality, 1-month readmissions, or 1-year mortality. Finally, in the multivariate analysis age, comorbidity indexes, with previous admissions, were independent predictors of 1-year mortality. The Authors conclude that different patterns of multimorbidity by gender, with greater functional impairment in women and more comorbidity in men exist, although without prognosis differences.

Overall, this is an interesting manuscript reporting on an intriguing topic. The aim is clear, and the analysis is well conducted and reported. I only have minor concerns about the role of clinical frailty, which the authors only mentioned in the abstract and in the introduction section, but did not explore as a variable in the analysis of comorbidities. Finally, the manuscript needs native English review and there are several typing errors.

For this reason, I don’t consider the manuscript suitable for publication as is stands.

Reviewer #2: The research aim was to evaluate differences in patterns of multimorbidity by gender in 843 patients admitted to a specific unit for chronic complex care patients and in-hospital mortality, 1-month readmissions, and 1-year mortality. Multimorbidity criteria, Charlson, PROFUND and Barthel indexes, and Pfeiffer test were collected prospectively during admission. 49.2% were men, with a median age of 84 years. Women were older, with greater functional dependence more cognitive deterioration and worse scores on the PROFUND index while men had more comorbidity measured with the Charlson index. In the multimorbidity criteria scale, heart failure, autoimmune diseases, dementia, and osteoarticular diseases were more frequent in women, while ischemic heart disease, chronic respiratory diseases, and neoplasms predominated in men. In the analysis of grouped patterns, neurological and osteoarticular diseases were more frequent in females, while respiratory and cancer predominated in males. We did not find gender differences for in-hospital mortality, 1- month readmissions, or 1-year mortality. In the multivariate analysis age, Charlson, Barthel and PROFUND indexes, alongside previous admissions, were independent predictors of 1-year mortality, while gender was non-significant. Charlson and PROFUND indexes predicted mortality during follow-up more accurately in men than in women (AUC 0.70 vs. 0.57 and 0.74 vs. 0.62), respectively, with both p<0.001.

The study is well conducted, data support conclusions. The manuscript is well written.

Figure 4 should be modified. Le legend is h and m and should be translated in English. The number of subject at risk should be described. The intervals should be 0 – 90 - 180 -270 – 360.

6. PLOS authors have the option to publish the peer review history of their article (what does this mean?). If published, this will include your full peer review and any attached files.

Reviewer #1: No

Reviewer #2: No

---

## [Author Response · Author response to Decision Letter 0]

2 Dec 2019

October 30, 2019

Pasquale Abete

Academic Editor

PLOS ONE

We thank you for assessing our manuscript entitled "Multimorbidity gender patterns in hospitalized elderly patients" which was submitted as an original article to PlosOne journal, and for giving us the opportunity to resubmit. We sincerely thank the PlosOne peer-reviewers for their constructive criticism and suggestions that indeed improved our original submission. To make this revision more reader-friendly, we have responded in red following each bullet with “RESPONSE:” Any new or modified text is also marked in red italics in the text. 

Point-by-point response addressing the queries and suggestions of associate editor and reviewers.

Editor Comments

The authors evaluate differences in patterns of multimorbidity by gender in this population and their possible prognostic implications defined as in-hospital mortality, 1-month readmissions, and 1- year mortality in 843 patients with well-established multimorbidity criteria admitted to a specific unit for chronic complex care patients. Multimorbidity criteria, Charlson, PROFUND and Barthel indexes, and Pfeiffer test were collected prospectively during admission. Women were older, with greater functional dependence, more cognitive deterioration, and worse scores on the PROFUND. In the multimorbidity criteria scale, heart failure, autoimmune diseases, dementia, and osteoarticular diseases were more frequent in women, while ischemic heart disease, chronic respiratory diseases, and neoplasms predominated in men. In the analysis of grouped patterns, neurological and osteoarticular diseases were more frequent in females, while respiratory and cancer predominated in males. We did not find gender differences for in-hospital mortality, 1- month readmissions, or 1-year mortality. In the multivariate analysis age, Charlson, Barthel and PROFUND indexes, alongside previous admissions, were independent predictors of 1-year mortality, while gender was non-significant.

The manuscript is very interesting. I have only a minor concern. No mention about the prevalence and incidence of syncope that is well known to influence mortality and disability in older adults. Do you have any data about this matter. Please see and discuss Ungar A et al. Two-year morbidity and mortality in elderly patients with syncope. Age Ageing. 2011 Nov;40(6):696-702.

RESPONSE

We sincerely appreciate the editor’s comments. Regrettably, the prevalence and incidence of syncope were not assessed in our study. 

In accordance with the editor’s suggestion we have added the following to the text:

“Finally, other useful predictors of mortality in elderly multimorbid patients, such as the history of previous syncope, were not analyzed in our study.”

We have added a new reference:

Ungar A, Galizia G, Morrione A, Mussi C, Noro G, Ghirelli L, Masotti G, Rengo F, Marchionni N, Abete P. Two-year morbidity and mortality in elderly patients with syncope. Age Ageing. 2011 Nov;40(6):696-702. doi: 10.1093/ageing/afr109. 

Reviewer 1 Comments

In the manuscript entitled Multimorbidity gender patterns in hospitalized elderly patients, Almagro P. and co-authors report on the assessment of differences in patterns of multimorbidity by gender in a cohort of elderly patients and their possible prognostic implications defined as in-hospital mortality, 1-month readmissions, and 1-year mortality. To this aim Authors focused on a cohort of 843 elderly patients in which, based on the multimorbidity criteria scale, heart failure, autoimmune diseases, dementia, and osteoarticular diseases were more frequent in women, and ischemic heart disease, chronic respiratory diseases, and neoplasms predominated in men. Then the Authors grouped multimorbidity patterns and found that neurological and osteoarticular diseases were more frequent in females, while respiratory and cancer predominated in males but didn’t find gender-related differences for in-hospital mortality, 1-month readmissions, or 1-year mortality. Finally, in the multivariate analysis age, comorbidity indexes, with previous admissions, were independent predictors of 1-year mortality. The Authors conclude that different patterns of multimorbidity by gender, with greater functional impairment in women and more comorbidity in men exist, although without prognosis differences.

Overall, this is an interesting manuscript reporting on an intriguing topic. The aim is clear, and the analysis is well conducted and reported. I only have minor concerns about the role of clinical frailty, which the authors only mentioned in the abstract and in the introduction section, but did not explore as a variable in the analysis of comorbidities. Finally, the manuscript needs native English review and there are several typing errors.

RESPONSE

We sincerely thank the reviewer for these suggestions.

We concur with the reviewer’s observation. We did not perform a formal frailty test, and for this reason, we did not include a frailty analysis in our results. Of note, we do not use the term ‘frail’ to define our population. We also agree that frailty, disability, and comorbidities are frequently related but formally distinct clinical entities. Nevertheless, in accordance with the current definition of frailty (a state of increased vulnerability to poor resolution of homeostasis after a stressor event, which increases the risk of adverse outcomes, including falls, delirium, and disability), we believe that the vast majority of our population cannot be classified as robust or prefrail elderly. (1,2)

In accordance with the reviewer’s suggestion we have replaced the term ‘frail elders’ in the abstract with the term ‘vulnerable elders’. In the rest of our previous manuscript, the term frail elderly was used on an additional 4 occasions, all of them in reference to previous publications.

Introduction

“Multimorbidity is associated with a lower quality of life, worse prognosis, and an increase in health expenditure. In addition, a subgroup of those with multimorbidity is acknowledged to require more complex health care: frail elders with several concomitant chronic diseases, repeated hospitalizations, and frequent ambulatory health care requirements” 

Reference Buja A. Multimorbidity patterns in high-need, high-cost elderly patients. PlosOne 2018. “Efforts in the US to better classify PCHCN have shown that this group mainly consists of frail elderly people or individuals with multimorbidity”.

Discussion

1. Previous studies have shown that 70 - 80% of patients that die in hospital are frail, older patients with multiple comorbidities admitted through emergency rooms. 

Reference Stewart K. Clinical Medicine 2016;16:530-4 

Those who die in hospital fall into three broad categories: 1) Frail, older patients with multiple comorbidities, admitted as emergencies, account for most deaths (70–80%).

1. Fried LP. Untangling the concepts of disability, frailty, and comorbidity: implications for improved targeting and care. J Gerontol A Biol Sci Med Sci 2004;59:255-63.

2. Clegg A. Frailty in elderly people. Lancet 2013;381: 752-762.

Finally, the manuscript needs native English review and there are several typing errors.

RESPONSE

The previous version of our manuscript was reviewed by a professional copy-editor (Tom Yohannan). The current version has been reviewed again by the same copy-editor.

Reviewer 2 Comments

Reviewer #2: The research aim was to evaluate differences in patterns of multimorbidity by gender in 843 patients admitted to a specific unit for chronic complex care patients and in-hospital mortality, 1-month readmissions, and 1-year mortality. Multimorbidity criteria, Charlson, PROFUND and Barthel indexes, and Pfeiffer test were collected prospectively during admission. 49.2% were men, with a median age of 84 years. Women were older, with greater functional dependence more cognitive deterioration and worse scores on the PROFUND index while men had more comorbidity measured with the Charlson index. In the multimorbidity criteria scale, heart failure, autoimmune diseases, dementia, and osteoarticular diseases were more frequent in women, while ischemic heart disease, chronic respiratory diseases, and neoplasms predominated in men. In the analysis of grouped patterns, neurological and osteoarticular diseases were more frequent in females, while respiratory and cancer predominated in males. We did not find gender differences for in-hospital mortality, 1- month readmissions, or 1-year mortality. In the multivariate analysis age, Charlson, Barthel and PROFUND indexes, alongside previous admissions, were independent predictors of 1-year mortality, while gender was non-significant. Charlson and PROFUND indexes predicted mortality during follow-up more accurately in men than in women (AUC 0.70 vs. 0.57 and 0.74 vs. 0.62), respectively, with both p<0.001.

The study is well conducted, data support conclusions. The manuscript is well written.

Figure 4 should be modified. Le legend is h and m and should be translated in English. The number of subject at risk should be described. The intervals should be 0 – 90 - 180 -270 – 360.

RESPONSE

We sincerely appreciate the opinions of the reviewer. Figure 4 has been revised in accordance with the reviewer's suggestions.

---

## [Decision Letter · Decision Letter 1]

17 Dec 2019

MULTIMORBIDITY GENDER PATTERNS IN HOSPITALIZED ELDERLY PATIENTS.

PONE-D-19-26678R1

Dear Dr. ALMAGRO,

We are pleased to inform you that your manuscript has been judged scientifically suitable for publication and will be formally accepted for publication once it complies with all outstanding technical requirements.

With kind regards,

Pasquale Abete

Academic Editor

PLOS ONE

Additional Editor Comments (optional):

No further comments.

Reviewers' comments:

Reviewer's Responses to Questions

**Comments to the Author**

1. If the authors have adequately addressed your comments raised in a previous round of review and you feel that this manuscript is now acceptable for publication, you may indicate that here to bypass the “Comments to the Author” section, enter your conflict of interest statement in the “Confidential to Editor” section, and submit your "Accept" recommendation.

Reviewer #1: All comments have been addressed

Reviewer #2: All comments have been addressed

2. Is the manuscript technically sound, and do the data support the conclusions?

Reviewer #1: Yes

Reviewer #2: Yes

3. Has the statistical analysis been performed appropriately and rigorously? 

Reviewer #1: Yes

Reviewer #2: Yes

4. Have the authors made all data underlying the findings in their manuscript fully available?

Reviewer #1: Yes

Reviewer #2: Yes

5. Is the manuscript presented in an intelligible fashion and written in standard English?

Reviewer #1: Yes

Reviewer #2: Yes

6. Review Comments to the Author

Reviewer #1: The authors have addressed all the comments. The manuscript is now suitable for publication in PLOS ONE.

Reviewer #2: All comments have been addressed. The manuscript is improved and in my opinion is suitable for publication..

7. PLOS authors have the option to publish the peer review history of their article (what does this mean?). If published, this will include your full peer review and any attached files.

Reviewer #1: No

Reviewer #2: No

---

## [Editor Report · Acceptance letter]

3 Jan 2020

PONE-D-19-26678R1 

MULTIMORBIDITY GENDER PATTERNS IN HOSPITALIZED ELDERLY PATIENTS. 

Dear Dr. Almagro:

I am pleased to inform you that your manuscript has been deemed suitable for publication in PLOS ONE. Congratulations! Your manuscript is now with our production department. 

With kind regards,

on behalf of

Prof. Pasquale Abete 

Academic Editor

PLOS ONE